# Associations between Sociodemographic, Dietary, and Substance Use Factors with Self-Reported 24-Hour Movement Behaviors in a Sample of Brazilian Adolescents

**DOI:** 10.3390/ijerph18052527

**Published:** 2021-03-04

**Authors:** Bruno Gonçalves Galdino da Costa, Jean-Philippe Chaput, Marcus Vinicius Veber Lopes, Luis Eduardo Argenta Malheiros, Kelly Samara da Silva

**Affiliations:** 1Núcleo de Pesquisa em Atividade Física e Saúde, Universidade Federal de Santa Catarina, Florianópolis 88040-900, Brazil; marcusvvl@hotmail.com (M.V.V.L.); luiseduardoam@gmail.com (L.E.A.M.); ksilvajp@gmail.com (K.S.d.S.); 2Healthy Active Living and Obesity Research Group, Children’s Hospital of Eastern Ontario Research Institute, Ottawa, ON K1H 8L1, Canada; jpchaput@cheo.on.ca

**Keywords:** physical activity, sedentary behavior, sleep, public health, 24 h movement behaviors

## Abstract

We aimed to identify sociodemographic, dietary, and substance use factors associated with self-reported sleep duration, physical activity (PA), and sedentary behavior (SB) indicators in a sample of Brazilian adolescents. Adolescents (*n* = 731, 51% female, mean age: 16.4 years) answered a questionnaire. The volume of total PA, sports, non-sports, total SB, leisure-time SB, involuntary SB, sleep duration, dietary behaviors, sociodemographic, and substance use indicators were self-reported. Multilevel linear models were fitted. Females engaged in less total PA, sports, total SB, and leisure-time SB, but in more involuntary SB than males. Age was positively associated with non-sports and involuntary SB. Socioeconomic status was positively associated with total PA. Adolescents who lived with the mother only practiced more sports compared to those living with two parents. Unprocessed food was positively associated with total PA and sports. Processed food was inversely associated with total PA and non-sports, and positively associated with total SB and leisure-time SB. Alcohol use was positively associated with total PA, and tobacco smoking was negatively associated with total PA. No associations were observed for sleep duration. In conclusion, sociodemographic, dietary, and substance use factors are associated with the 24 h movement behaviors among Brazilian adolescents, and some associations are type specific.

## 1. Introduction

The 24 h movement behaviors, composed of sleep duration, sedentary behavior, and physical activity, are associated with several health outcomes among adolescents [1], and engaging in adequate volumes of each behavior has been shown to provide multiple health-related benefits during adolescence [2,3,4,5,6]. The importance of these behaviors has been highlighted in integrated 24 h movement recommendations [7], and policies and interventions have aimed to increase physical activity [8], decrease sedentary behaviors [9], and promote adequate sleep [10] among adolescents and reported modest changes [8,9,10]. Adolescents’ engagement in physical activity is low [11,12], a large proportion of their day is spent sedentary [13], and inadequate sleep is common [14], culminating in unhealthy patterns of the 24 h movement behaviors [1]. In fact, the adherence of adolescents with the 24 h movement behavior guidelines has been very low, ranging from 1.6% to 9.7% [1], and intrapersonal factors such as sociodemographic, dietary, and substance use indicators can be related to guideline adherence [15,16,17,18].

Previous studies have shown that females are less active [15], sleep more [19], and commonly engage in less sedentary behavior than males [13]. Age has also been negatively associated with physical activity [15] and sleep [20], and positively related to sedentary behavior in adolescents [21], while socioeconomic status (SES) is related to these behaviors depending on the country setting, and the domain of physical activity [22,23]. Family structure has also been shown to play a role in some behaviors, with conflicting evidence pointing to either single-parent [24] or two-parent [25] families having more active children. Dietary behavior may also play a role, as it has been associated with physical activity [16,26], sedentary behavior [16,17], and sleep [27] in previous studies. Besides that, diet impacts physiological mechanisms related to hormones and the availability of nutrients, which may be associated with the 24 h movement behaviors [28,29]. Alcohol use, tobacco smoking [18], and, with less evidence, illicit drug experimentation [30] have also been associated with physical activity [18,30], sedentary behavior [18], and sleep [31]. Overall, lifestyle behaviors and sociodemographic characteristics seem to be linked to the 24 h movement behaviors, but there are only a few studies available on the topic [15,16,30], and most of them come from high-income countries, which may not be applicable to lower- and middle-income settings [32].

In addition to the relationship between the volume (amount) of physical activity and sedentary behavior, some sociodemographic, dietary, and substance use factors may be related to specific types of physical activity and sedentary behaviors [33,34]. For example, alcohol use has been linked specifically with participation in sports among adolescents in a review of longitudinal studies [30]. This relationship may not be observed for other physical activities, such as commuting or doing chores. Drawing a parallel with sedentary behavior, unhealthy snacking has been linked specifically to increased television viewing [29], but may not be observed with mandatory or involuntary screen time such as the time spent studying or working with screen-based devices. Specific physical activities and sedentary behaviors affect health differently, with sports, for example, being more beneficial for specific dimensions of health compared to non-sport activities among adolescents [5]. This is also observed for sedentary behaviors, with some specific activities such as using social media being related to unhealthy outcomes like depressive symptoms, whereas studying is not [6]. Identifying factors associated with specific subtypes of physical activity and sedentary behavior may be helpful, as some subgroups such as females may need specific interventions to engage in more sports [5], while males may need specific strategies to reduce leisure-time sedentary behaviors [13].

Considering that most adolescents do not simultaneously engage in healthy levels of physical activity, sedentary behavior, and sleep duration [1], identifying correlates of these behaviors is desirable for research, practice, and policy. Furthermore, investigating several sociodemographic and lifestyle behaviors in a middle-income setting is desirable to broaden the understanding of how adolescents from these countries behave, and how these behaviors are linked, which is not consistently depicted in the scientific literature. Thus, the present study aimed to identify sociodemographic, dietary, and substance use factors associated with self-reported sleep duration, physical activity, and sedentary behavior indicators in a sample of Brazilian adolescents.

## 2. Materials and Methods

### 2.1. Participants

This study used data from adolescents aged 14–18 years that participated in the baseline of the ELEVA project (Estudo Longitudinal do Estilo de Vida de Adolescentes/Longitudinal Study of the Lifestyle of Adolescents), conducted between August and December 2019. All three public schools offering professional courses integrated with high schools in the mesoregion Grande Florianópolis, southern Brazil, were invited and accepted to participate in the project. Each school was located in one city of the mesoregion (Florianópolis, São José, and Palhoça). Florianópolis is the capital of Santa Catarina State, inside the mesoregion Grande Florianópolis. The municipality has 421,000 inhabitants, and a human development index (HDI) of 0.847, the third highest of the country in 2010, and the highest amongst the Brazilian state capitals. The Gini index (the closer to zero, the less social inequality) was 0.54 in 2010 [1]. São José is, with 209,000 inhabitants and an HDI of 0.809, in the very high stratum (above 0.8), and has a Gini index of 0.44 [1]. Palhoça is a city in Santa Catarina State, inside the mesoregion Grande Florianópolis. The municipality has 137,000 inhabitants and an HDI of 0.757, with a Gini index of 0.40 [1]. A census method was adopted, and all high school students who did not have any disability that prevented them from participating in the study were invited to participate. A total of 1618 students were enlisted in the schools, of whom 1269 were present in classes during the data collection period and were invited to participate. The study was explained by trained researchers during school hours, and informed consent forms were given to students for their parents or legal guardians to sign, and an assent form was given to all students to sign. A total of 1010 students provided written informed consent and participated in the study. The present research project was approved by the Ethics Committee in Research with Human Beings of the Universidade Federal de Santa Catarina (protocol number: 3.168.745).

### 2.2. Measures

Participants answered a standardized survey, hosted on an online platform, using smartphones, tablets, and/or computers. Students could use either their device and internet connection, but devices and wireless internet were also offered by the research team. The average time to complete the survey was 24 min.

#### 2.2.1. 24 h Movement Behaviors

Physical activity was assessed using an adapted version of the self-assessed physical activity checklist [35]. The questionnaire is composed of the following question: *In general, which of the following activities do you practice? Report how many days in a typical week and for how long each a day do you engage in any of the activities*, followed by a list of 22 activities, where participants reported the weekly frequency (0–7) and the duration in minutes of each session of each activity. Three variables were calculated: the total volume of physical activity (sum of the volume of each activity), the volume of sports (comprising soccer, futsal, basketball, handball, volleyball, tennis, table tennis, swimming, athletics, combat sports, gymnastics, cycling, skating, and surfing), and the volume of non-sport activities (comprising capoeira, dancing, collective gymnastics and gym going, weight lifting, walking, jogging, and active play). This questionnaire was validated for Brazilian adolescents [36], and the classification of sports and non-sport activities was previously used in research with adolescents [5,6].

Sedentary behavior was measured by a proxy of screen time across four variables: time spent studying, working, watching videos (e.g., series, news, sports, streams, and movies), and playing video games using screen-based devices. Participants reported the time in hours and minutes (in blocks of 10 min) they engaged per day in each of those activities during weekdays (Mondays to Fridays), and weekend days (Saturdays and Sundays). This questionnaire was developed and validated for Brazilian adolescents, and further detail is available elsewhere [37]. Daily screen time was calculated by weighting the volume on weekends by 2 and on weekdays by 5. Three variables were calculated: the total volume of sedentary behavior, by summing the daily volume of the four indicators; the involuntary sedentary behavior, by summing time spent studying and working; and the leisure-time sedentary behavior, by summing the time spent watching videos and playing video games.

Finally, participants reported the time they habitually went to bed and woke up on weekdays and weekend days, and sleep duration was calculated as the difference between both. These questions have been used in research with Brazilian adolescents [38]. The habitual sleep duration was calculated by weighting the sleep duration on weekends by 2 and on weekdays by 5 and dividing the resulting value by 7.

#### 2.2.2. Dietary Patterns

The weekly frequency of consumption of beans, vegetables, fresh fruits, salted fried food, sweets, soda, ultra-processed sausages, and fast food was reported by participants. This questionnaire was previously validated for Brazilian adolescents [39]. For each item, answers ranging from 0–7 days/week were used to create two scores using confirmatory factor analyses. The first score was named “Unprocessed food”, comprising the indicators of vegetables and fresh fruits, and the second score was named ‘Processed food’, and calculated based on salted fried food, sweets, soda, ultra-processed sausages, and fast food. The standardized scores were calculated using standardized variables, but for ease of interpretation, they were scaled from 0 to 10, where 10 means the highest consumption of each food group.

#### 2.2.3. Substance Use

Alcohol use and tobacco smoking were reported for the last 30 days before data collection. The indicator variable was coded as a dichotomous variable, with participants being classified as smokers (vs. non-smokers) and drinkers (vs. non-drinkers) if they answered smoking and/or drinking at least once in the 30-day window before data collection. For illicit drug experimentation, participants reported if they had used any illicit drug in their lifetime (yes or no). These questions have been previously used in Pesquisa Nacional de Saúde do Escola (PeNSE) [40].

#### 2.2.4. Sociodemographic Variables

Participants reported their sex (male/female), age (completed years), family structure, and socioeconomic status (SES). The family structure indicator had four possible categories: living with either mother, father, both, or neither. For the SES indicator, participants reported the number (0–4 or more) for each of several household belongings (e.g., television, computer, fridges, stoves), if they had treated water, if the street where they lived was paved, and the highest education of their parents. Each indicator (number of household belongings, treated water, pavement, and parental education) received a weight based on the responses (e.g., ownership of four or more cars weights more than four or more fridges), and were used to calculate a nationally standardized socioeconomic score that ranged from 0 to 100, with 100 being the highest SES. The measurement of the indicators and the calculation of the score were conducted according to the guidelines of the Brazilian Association of Research Companies [41].

### 2.3. Statistical Analyses

The characteristics of the sample are described using means and standard deviations, and absolute and relative frequencies. To test the associations between sociodemographic (sex, age, SES, family structure), dietary (unprocessed food score, processed food score), and substance use (alcohol use, tobacco use, illicit drug experimentation) with sleep duration, total sedentary behavior, leisure-time sedentary behavior, and involuntary sedentary behavior, multilevel linear regression models were estimated using a restricted maximum likelihood approach, and 95% confidence intervals (CIs) were also estimated. For total physical activity, sports, and non-sports physical activity, generalized linear multilevel models with the gamma family were fit using the adaptive Gauss–Hermite guadrature, and the 95% CI was calculated. This was used as all physical activity indicators were right-skewed, and the gamma models are recommended for such distributions and provided better residuals [42]. All models were mutually adjusted for all independent variables, and a random intercept for each school was estimated to take into account the hierarchical structure of the data, with students nested within schools. Statistical significance was set to *p*-value < 0.05 (two-tailed). The analyses were conducted using R, version 4.0 for Windows (R Foundation for Statistical Computing, Vienna, Austria), using the lme4 package.

## 3. Results

Of the total of 1010 participants who provided written informed consent, 731 (72%) had complete data in all survey variables and were included in the present analyses. The characteristics of the participants can be observed in Table 1.

The association between sociodemographic, dietary, and substance use factors with physical activity, sports, and non-sports physical activity are displayed in Table 2, and results are presented as mean differences (β). Females engaged in less total physical activity (β = −16.75, 95%CI −22.12; −11.37; i.e., on average, females engaged in 16.75 fewer minutes of total physical activity than males) and sports (β = −20.34, 95%CI −25.32; −15.36) compared to males, while a positive relationship was observed between age and non-sports physical activity (β = 3.25, 95%CI 1.43; 5.06). The SES score was positively associated with total physical activity (β = 0.30, 95%CI 0.07; 0.53), and family structure was associated with sports, with participants living with their mother engaging in more sports than those living with both parents (β = 3.25, 95%CI 0.16; 6.33). Unprocessed food was positively related to total physical activity (β = 2.06, 95%CI 1.16; 2.96) and sports (β = 0.92, 95%CI 0.49; 1.35), while processed food was inversely related to total physical activity (β = −1.91, 95%CI −3.01; −0.82) and non-sports physical activity (β = −1.03, 95%CI −1.98; −0.08). Alcohol intake was positively associated with total physical activity (β = 5.95, 95%CI 1.05; 10.84), and tobacco use was negatively related to total physical activity (β= −11.54, 95%CI −17.13; −5.96). Illicit drug experimentation was not significantly associated with any physical activity indicator.

The association between sociodemographic, dietary, and substance use factors with sedentary behavior indicators can be observed in Table 3. Females spent less time in total (β = −1.21, 95%CI −1.80; −0.65) and leisure-time sedentary behavior (β = −1.87, 95%CI −2.32; −1.42), but more time in involuntary (β = 0.61, 95%CI 0.28; 0.94) sedentary behavior than males. Age was positively associated with involuntary sedentary behavior (β = −0.27, 95%CI 0.11; 0.43). Processed food was positively related to total (β= 0.45, 95%CI 0.28; 0.62) and leisure-time (β= 0.36, 95%CI 0.23; 0.49) sedentary behavior. SES, family structure, unprocessed food score, tobacco use, alcohol consumption, and illicit drug experimentation were not significantly associated with sedentary behavior indicators.

Table 4 shows that the sociodemographic, dietary, and substance use factors were not significantly associated with sleep duration in the present study.

## 4. Discussion

The aim of the current study was to evaluate the associations between sociodemographic, dietary, and substance use factors with self-reported sleep duration, physical activity, and sedentary behavior indicators in a sample of Brazilian adolescents. In general, we found that the relationships between sociodemographic, dietary, and substance use factors with physical activity and sedentary behavior are type specific, and no associations were observed for sleep.

Our findings show that females engage in less total physical activity and sports, but not non-sport activities, compared to males. A review showed that females are usually less active than males in most scientific studies [15]; however, this may also be dependent on the way physical activity is measured, as some evidence shows that this relationship is not true for all domains of physical activity [33]. Non-sport activities include activities such as walking and engaging in gymnastics, which are typically preferred by females [43], whereas males may opt to engage in more sports due to their preference for this kind of leisure activity. Evidence shows that males overall prefer practicing physical activity over other leisure time activities, and that they enjoy practicing physical activity more compared to their female counterparts [44].

The results of our study also show that similar associations were observed for sedentary behaviors, as females engaged in less total and leisure sedentary behaviors, but more in involuntary sedentary behaviors compared to males. When domain-specific sedentary behaviors were analyzed in a sample of 9218 adolescents and adults from eight Latin American countries, associations between sex and sedentary activities were domain dependent, and in some cases country dependent [45]. For example, males engaged in more total sedentary behavior than females, but no differences were found for reading or talking on the phone, and video game use was higher for males in some countries but not others [45]. One reason as to why females in our study engaged in more involuntary sedentary activities than males relates to a higher volume of time spent studying, but may also be due to the fact that they engage in more sedentary work-related activities compared to males. These hypotheses have to be further explored in future studies. Further, limiting involuntary sedentary behaviors may not always be an option, as working and studying using screens may be required even more with the advancement of technologies and applications.

In relation to age and family structure, no associations were observed for total physical activity, total sedentary behavior, or sleep duration. However, age was positively associated with non-sports activities and involuntary sedentary behaviors. This may be explained by increased autonomy and responsibilities given to adolescents with age, as they join the workforce and may have increased loads of activities at school, while also being able to actively commute more safely. Similar trends were observed in Chinese children and adolescents between 2004 and 2011 [33], showing an increase in domestic physical activity and computer use. As for SES, a positive relation was observed for total physical activity, which may be associated with increased opportunities to buy equipment (e.g., bicycles) and afford clubs or gyms; however, as this association was not observed for either the non-sport or sport variable, this relation is not entirely clear. The association between SES and physical activity has been highlighted in previous studies, but the direction of the association may not be the same across countries [32], although evidence suggests that higher SES is associated with higher physical activity in Latin American countries [34]. Lastly, family structure was associated with sports, with adolescents living only with their mothers reporting spending more time practicing sports compared to their peers living with both their father and mother. This finding is in line with results observed in a study with Chinese adolescents, where those who lived with single parents were more active than those living in two-parent homes [24]. However, a study involving Norwegian adolescents observed that single-parent families were unfavorable for physical activity and sport participation compared to two-parent homes [25]. A hypothesis is that adolescents in single-parent families may have more unsupervised time and autonomy to engage in other activities such as sports. However, it is difficult to determine how the family structure impacts behaviors such as physical activity, as it may be determined by social and cultural factors that vary between nations.

Our results also indicate significant associations between dietary behaviors and physical activity and sedentary behavior, but not with sleep duration. Adolescents who reported eating more unprocessed foods had higher physical activity and sports participation, which may be attributed to a preference for healthy behaviors. Previous studies have shown that these behaviors tend to cluster among adolescents [16]. This corroborates the positive associations observed in the present study between processed food scores and sedentary behaviors, and the negative association of this score with physical activity indicators. This is supported by a recent review that found that lower vegetable and fruit consumption, and higher consumption of energy-dense snacks and sugar-sweetened beverages were associated with higher levels of sedentary behavior in pediatric populations [17]. Snacking and unhealthy eating have been associated with time watching television [29] and playing video games [28], which may be explained by mental stress and cognitive processes related to reward systems [28]. Additionally, experimental studies have shown that eating while watching television may cause a delay in satiation and a reduction in internal satiety signals [46,47]. Another possible explanation is that depending on the content and quantity of meals, postprandial sleepiness may occur, and adolescents may opt to engage in sedentary activities rather than physical activities if they are feeling tired or somnolent [48].

In relation to substance use, associations between alcohol and tobacco use and physical activity were observed, but not for sleep or sedentary behaviors. While a positive relationship was observed between alcohol and total physical activity, no associations were observed for either sports or non-sports. The use of alcohol has been positively associated with physical activity among adolescents in other studies [30], and this relationship has been at least partially attributed to sport participation, as increased social activities after practice or during competitions would serve as opportunities for adolescents to experiment and drink alcoholic beverages. However, our results do not support this hypothesis, as we have not observed a significant association between sports and alcohol intake. It is not clear why this relationship was observed, and future studies should include more detailed questions in longitudinal designs to clarify this finding. With regard to tobacco, a negative association with total physical activity was observed, which has also been observed among youth before [49]. Independently of the modality, practicing physical activity is usually associated with a healthier lifestyle, which would also predispose participants to not engage in smoking. In relation to illicit drugs, we have observed no significant associations. A review has shown that results of this association are mixed [30], with varying results regarding experimentation and usage, and by type of drug [30].

There are several strengths in the present study, including the use of several independent variables that are not commonly analyzed together in studies with adolescents. Secondly, the inclusion of subtypes of physical activities and sedentary behaviors can provide additional useful information that cannot be obtained with the use of device-based measures. Thirdly, the inclusion of a sample of adolescents living in a middle-income setting is worthy of note, as most of the evidence in this field is from studies conducted in high-income countries. This study also has limitations that are important to mention, such as the cross-sectional design, which precludes drawing cause-and-effect associations. The use of self-reported instruments, although necessary for the identification of types of activities, may be prone to memory and social desirability biases (e.g., not admitting drinking or smoking, as the legal age for such behaviors is 18 years in Brazil). Lastly, we have not included a non-screen-based sedentary behavior indicator, which may have underestimated sedentary behaviors.

## 5. Conclusions

In conclusion, we found that the association between sociodemographic, dietary, and substance use factors with the 24 h movement behaviors not only depends on the volume of the behavior, but also on its type. We found that males engaged in more sports and leisure-time screen time, and less involuntary screen time (i.e., working and studying) than females. Older individuals engaged in more non-sport physical activities and involuntary screen time, and individuals with higher SES engaged in more total physical activity. Higher ingestion of processed food was associated with less total physical activity and increased leisure-time and total screen time, whereas increased consumption of non-processed food was associated with higher participation in sports and total physical activity. Smoking was associated with lower total physical activity, but alcohol and illicit drugs were not associated with any 24 h movement behavior. Finally, no significant associations were found for sleep duration. These results are relevant for public health, given the importance of engaging in healthy levels of physical activity, sedentary behavior, and sleep. The inclusion of specific sedentary behavior and physical activity types is also novel, as they have different correlates and relate to health in different ways. Efforts should be made to promote physical activity, especially among females, older adolescents, and those at risk of socioeconomic vulnerability, while screen time should also be decreased, especially among males. Promoting healthy eating may also support changes in physical activity and screen time. Future studies should confirm these associations and determine the directionality using longitudinal designs, and these factors should be taken into consideration when planning interventions and policies among adolescents.

## Figures and Tables

**Table 1 ijerph-18-02527-t001:** Characteristics of the sample (mean ± SD or *n* (%)).

	Total Sample (*n* = 731)	Females (*n* = 375)	Males (*n* = 356)
Age (years)	16.37 ± 1.04	16.35 ± 1.04	16.38 ± 1.04
SES score (0–100)	48.97 ± 9.98	48.33 ± 9.66	49.63 ± 10.28
Family structure [*n* (%)]			
Live with both parents	467 (63.9)	234 (62.4)	233 (65.4)
Live with mother	195 (26.7)	104 (27.8)	91 (25.6)
Live with father	30 (4.1)	17 (4.5)	13 (3.7)
Other	39 (5.3)	20 (5.3)	19 (5.3)
Dietary behavior			
Unprocessed food (0–10)	5.93 ± 2.68	6.20 ± 2.74	5.66 ± 2.59
Processed food (0–10)	3.76 ± 1.86	3.66 ± 1.88	3.86 ± 1.82
Alcohol use in the last 30 days [*n* (%)]			
No	415 (56.8)	207 (55.2)	208 (58.4)
Yes	316 (43.2)	168 (44.8)	148 (41.6)
Tobacco use in the last 30 days [*n* (%)]			
No	677 (92.6)	344 (91.7)	333 (93.5)
Yes	54 (7.4)	31 (8.3)	23 (6.5)
Illicit drug experimentation [*n* (%)]			
No	594 (81.3)	286 (76.3)	308 (86.5)
Yes	137 (18.7)	89 (23.7)	48 (13.5)
Sleep duration (hours/night)	8.02 ± 1.30	8.05 ± 1.26	7.98 ± 1.34
Physical activity (minutes/day)			
Total physical activity	37.30 ± 37.44	30.90 ± 34.11	44.03 ± 39.60
Sports	18.27 ± 31.06	9.48 ± 18.40	27.53 ± 38.20
Non-sports	20.75 ±31.18	21.67 ± 30.80	19.79 ± 31.58
Sedentary behavior (hours/day)			
Total sedentary behavior	7.56 ± 4.06	6.90 ± 3.82	8.25 ± 4.19
Leisure-time sedentary behavior	4.68 ± 3.30	3.72 ± 2.79	5.69 ± 3.50
Involuntary sedentary behavior	2.76 ± 2.29	3.05 ± 2.34	2.45 ± 2.19

SES: socioeconomic status; SD: standard deviation.

**Table 2 ijerph-18-02527-t002:** Association between sociodemographic, dietary, and substance use factors with physical activity, sports, and non-sport practice in a sample of Brazilian adolescents (*n* = 731).

	Total Physical Activity (min/Day)	Sports (min/Day)	Non-Sports (min/Day)
	Coefficient (95%CI)	Coefficient (95%CI)	Coefficient (95%CI)
Sex			
Male	Reference	Reference	Reference
Female	**−16.75 (−22.12; −11.37)**	**−20.34 (−25.32; −15.36)**	0.83 (−2.53; 4.18)
Age (years)	−0.61 (−2.40; 1.18)	0.44 (−0.57; 1.45)	**3.25 (1.43; 5.06)**
SES score (0–100)	**0.30 (0.07; 0.53)**	0.00 (−0.10; 0.10)	0.02 (−0.15; 0.19)
Family structure			
Live with both parents	Reference	Reference	Reference
Live with mother	5.51 (0.41; 10.61)	**3.25 (0.16; 6.33)**	−0.50 (−5.15; 4.14)
Live with father	4.66 (−7.87; 17.19)	0.08 (−3.00; 3.15)	−1.26 (−8.52; 6.00)
Other	0.85 (−5.16; 6.85)	2.73 (−0.90; 6.36)	0.26 (−7.48; 8.01)
Dietary behavior			
Unprocessed food (0–10)	**2.06 (1.16; 2.96)**	**0.92 (0.49; 1.35)**	0.33 (−0.35; 1.00)
Processed food (0–10)	**−1.91 (−3.01; −0.82)**	0.42 (−0.18; 1.02)	**−1.03 (−1.98; −0.08)**
Alcohol use in the last 30 days			
No	Reference	Reference	Reference
Yes	**5.95 (1.05; 10.84)**	1.68 (−0.46; 3.81)	−0.94 (−4.7; 2.83)
Tobacco use in the last 30 days			
No	Reference	Reference	Reference
Yes	**−11.54 (−17.13; −5.96)**	−2.40 (−5.61; 0.81)	5.36 (−2.88; 13.60)
Illicit drug experimentation			
No	Reference	Reference	Reference
Yes	−0.80 (−6.67; 5.07)	0.39 (−2.71; 3.49)	3.50 (−2.33; 9.34)

Associations were tested using generalized linear multilevel models using the gamma family. 95%CI: 95% confidence interval; SES: socioeconomic status. Bold values refer to significant associations at *p* < 0.05.

**Table 3 ijerph-18-02527-t003:** Association between sociodemographic, dietary, and substance use factors with total, leisure, and involuntary sedentary behavior in a sample of Brazilian adolescents (*n* = 731).

	Total Sedentary Behavior (min/Day)	Leisure-Time Sedentary Behavior (min/Day)	Involuntary Sedentary Behavior (min/Day)
	Coefficient (95%CI)	Coefficient (95%CI)	Coefficient (95%CI)
Sex			
Male	Reference	Reference	Reference
Female	**−1.21 (−1.80; −0.65)**	**−1.87 (−2.32; −1.42)**	**0.61 (0.28; 0.94)**
Age (years)	0.22 (−0.06; 0.50)	−0.08 (−0.30; 0.13)	**0.27 (0.11; 0.43)**
SES score (0–100)	0.02 (−0.01; 0.05)	0.01 (−0.01; 0.04)	0.00 (−0.02; 0.02)
Family structure			
Live with both parents	Reference	Reference	Reference
Live with mother	0.28 (−0.39; 0.96)	0.27 (−0.26; 0.8)	−0.04 (−0.42; 0.35)
Live with father	0.60 (−0.85; 2.05)	−0.20 (−1.33; 0.94)	0.64 (−0.19; 1.48)
Other	0.22 (−1.07; 1.51)	0.11 (−0.90; 1.12)	−0.05 (−0.79; 0.69)
Dietary behavior			
Unprocessed food (0–10)	−0.05 (−0.17; 0.07)	−0.05 (−0.15; 0.04)	0.02 (−0.05; 0.08)
Processed food (0–10)	**0.45 (0.28; 0.62)**	**0.36 (0.23; 0.49)**	0.08 (−0.01; 0.18)
Alcohol use in the last 30 days			
No	Reference	Reference	Reference
Yes	−0.22 (−0.83; 0.39)	0.10 (−0.38; 0.58)	−0.33 (−0.69; 0.02)
Tobacco use in the last 30 days			
No	Reference	Reference	Reference
Yes	−0.34 (−1.58; 0.86)	0.52 (−0.45; 1.46)	−0.79 (−1.49; −0.09)
Illicit drug experimentation			
No	Reference	Reference	Reference
Yes	0.43 (−0.44; 1.29)	0.14 (−0.54; 0.81)	0.31 (−0.19; 0.80)

Associations were tested using linear multilevel regression models. 95%CI: 95% confidence interval; SES: socioeconomic status. Bold values refer to significant associations at *p* < 0.05.

**Table 4 ijerph-18-02527-t004:** Association between sociodemographic, dietary, and substance use factors with sleep duration in a sample of Brazilian adolescents (*n* = 731).

	Sleep Duration (h/Night)
	Coefficient (95%CI)
Sex	
Male	Reference
Female	0.10 (−0.10; 0.29)
Age (years)	−0.08 (−0.17; 0.01)
SES score (0–100)	0.00 (−0.01; 0.01)
Family structure	
Live with both parents	Reference
Live with mother	0.07 (−0.15; 0.29)
Live with father	−0.05 (−0.52; 0.43)
Other	−0.26 (−0.68; 0.17)
Dietary behavior	
Unprocessed food (0–10)	−0.03 (−0.07; 0.01)
Processed food (0–10)	−0.05 (−0.10; 0.01)
Alcohol use in the last 30 days	
No	Reference
Yes	−0.16 (−0.36; 0.04)
Tobacco use in the last 30 days	
No	Reference
Yes	−0.06 (−0.46; 0.34)
Illicit drug experimentation	
No	Reference
Yes	−0.04 (−0.32; 0.25)

Associations were tested using linear multilevel regression models. 95%CI: profile 95% confidence interval; SES: socioeconomic status.

## Data Availability

The participants’ individual data are not publicly available as the original approval by the Ethics Committee of research with human beings of the Universidade Federal de Santa Catarina (protocol number: 3.168.745) and the informed consent from the participants and their respective legal guardians stated the individual information will not be shared.

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
