# Peer review of "Associations between Sociodemographic, Dietary, and Substance Use Factors with Self-Reported 24-Hour Movement Behaviors in a Sample of Brazilian Adolescents"

_ijerph, 2021, doi:10.3390/ijerph18052527_

Round 1

Reviewer 1 Report

This study examined that sociodemographic, dietary and substance use factors associated with self-reported sleep duration, physical activity, and sedentary behavior indicators in a sample of Brazilian adolescents. The authors found that sex, age, and dietary behaviors were associated with physical activity and sedentary behaviors, while SES, family structure, use of alcohol, and use of tobacco were associated only with physical activity. However, no significant associations were found for sleep duration. This study provides useful information for adolescent health. However, I have some concerns.

Participants

This study used data from adolescents aged 14-18 years from 3 public high schools in Brazil. However, the details of these adolescents were not explained enough (socioeconomic status, the difference between schools, and distribution of participants in 3 schools). Please explain the details of the study participants.

Sedentary behavior and sleep duration

The questionnaire of physical activity was validated in a previous study. However, sedentary behavior and sleep duration might not be assessed correctly.

Alcohol use and tobacco smoking

The legal drinking and smoking age in Brazil is 18 years old. Therefore, study participants might not answer honestly.

Socioeconomic status

Please explain the details of the indicators of socioeconomic status.

Author Response

Reviewer 1

  1. This study used data from adolescents aged 14-18 years from 3 public high schools in Brazil. However, the details of these adolescents were not explained enough (socioeconomic status, the difference between schools, and distribution of participants in 3 schools). Please explain the details of the study participants.

Response: Each school was situated in one city of the Mesoregion Grande Florianópolis, in southern Brazil. Information about each city (Florianópolis, São José, Palhoça) can contextualize the schools better. Florianópolis is the capital of the Santa Catarina State, inside the mesoregion Grande Florianópolis. The municipality has 421,000 inhabitants, and a Human Development Index (HDI) of 0.847, the third-highest of the country in 2010, and the highest amongst the Brazilian State capitals. The Gini Index (the closer to zero, there is the less social inequality) was of 0.54 in 2010 [1]. São José is, with 209,000 inhabitants, an HDI of 0.809, in the very high stratum (above 0,8), and a Gini index of 0.44 [1]. Palhoça is a city in the Santa Catarina State, inside the mesoregion Grande Florianópolis. The municipality has 137,000 inhabitants and an HDI of 0.757, with a Gini index of 0.40 [1].  

Briefly, the characteristics of the participants per school were:

  1. Florianópolis: n = 414; average age: 16.41 ±1.08 years; % of females: 53.4%
  2. São José: n = 171; average age: 16.30 ±0.98 years; % of females: 33.3;
  3. Palhoça: n = 146, average age: 16.32 ±1.00 years; % of females: 66.4.

Distinct characteristics were expected as the schools represent the context of each city. These variables were included as covariates in the adjusted models, and the school was included as a random intercept to address these differences.

We have included the descriptions of the population size, HDI, and Gini Index of the cities in the methods section of the manuscript, Lines 82-89.

  1. The questionnaire of physical activity was validated in a previous study. However, sedentary behavior and sleep duration might not be assessed correctly.

Response: The questionnaire used to assess sedentary behavior has been validated for Brazilian adolescents. More information has been provided, and a reference that describes the conceptualization and validation procedures in detail has been added.[2]

As for the self-reported sleep duration, we have adopted the same procedure used by other studies with adolescents. [3–5] These questions are based on the same structure of questions 1 and 3 of the Pittsburgh Sleep Quality Index (PSQI) to calculate sleep duration.[6]

A sentence providing information about the questions has been inserted in each paragraph, and references are also provided. Changes were made on lines 117-118 and 124-126.

‘This questionnaire was developed and validated for Brazilian adolescents, and further detail is available elsewhere.[37]’

‘This questionnaire was developed and validated for Brazilian adolescents, and further detail is available elsewhere.[37]’

‘These questions have been used in research with Brazilian adolescents.[38] The habitual sleep duration was calculated by weighting the sleep duration on weekends by 2 and on weekdays by 5, and dividing the resulting value by 7.’

  1. The legal drinking and smoking age in Brazil is 18 years old. Therefore, study participants might not answer honestly.

Response: We have stressed the possibility of dishonesty regarding these questions in the limitations paragraph. Changes were made on lines 283-284.

‘The use of self-reported instruments, although necessary for the identification of types of activities, may be prone to memory and social desirability biases (e.g., not admitting drinking or smoking, as the legal age for such behaviors is 18 years in Brazil).’

  1. Please explain the details of the indicators of socioeconomic status.

Response: We have expanded the paragraph explaining how the socioeconomic status variable was calculated, and how its indicators were measured.

‘For the SES indicator, participants reported the number (0-4 or more) for each of several household belongings (e.g., television, computer, fridges, stoves), if they had treated water, if the street where they lived was paved, and the highest education of their parents. Each indicator (number of household belongings, treated water, pavement, and parental education) received a weight based on the responses (e.g., ownership of four or more cars weights more than four or more fridges), and were used to calculate a nationally standardized socioeconomic score that ranged from 0 to 100, with 100 being the highest SES. The measurement of the indicators and the calculation of the score were conducted according to the guidelines of the Brazilian Association of Research Companies.[41] ‘

Changes were made on lines 146-153.

Reviewer 2 Report

The literature review is brief. It must be expanded.

The “Participants and measures” section is correct.

The "Result" section should work a little more and change certain aspects:

- It is missing  the level of significance used in the-value

- For the subject of table format, please review this article and change the format and way of presenting it (Int. J. Environ. Res. Public Health 2011, 8 (10), 3953-3978; doi.org/10.3390/ijerph8103953)

- It's not necessary to use in the time "95% CI"

- Throughout the text use the notation β = -16.75, 95% CI -22.12; -11.37. It should be explained the first time it appears in the text.

- Regression table 2, 3, and 4  I would not put the row that is "Reference", it's obviously

The "Discussion" section is correct

The "conclusion" section too short. It must be expanded.

Author Response

Reviewer 2

  1. The literature review is brief. It must be expanded.

Response: We have expanded the rationale regarding the activity subtypes in the third paragraph of the introduction. We believe this is the main novelty of the study, and the new sentences highlight the importance of identifying these activity subtypes and possible implications for public health.

Changes were made on lines 62-66.

‘Specific physical activities and sedentary behaviors affect health differently, where sports for example being more beneficial for specific dimensions of health compared to non-sport activities among adolescents. [5] This is also observed for sedentary behaviors, with some specific activities such as using social media being related to unhealthy outcomes like depressive symptoms whereas studying is not.[6]’

  1. The “Participants and measures” section is correct.

Response: Additional information has been provided to improve these sections further.

  1. The "Result" section should work a little more and change certain aspects:
    1. - It is missing the level of significance used in the-value

Response: The level of significance was illustrated by the use of bold text on the tables. Every association where the value of p < 0.05 was highlighted with bold text. This has been described in the footnotes of tables 2 and 3. The following sentence was included in line 166 in the Methods section: “Statistical significance was set to p-value <0.05 (two-tailed)”.

  1. - For the subject of table format, please review this article and change the format and way of presenting it (Int. J. Environ. Res. Public Health 2011, 8 (10), 3953-3978; doi.org/10.3390/ijerph8103953)

Response: It is not clear which aspects of the table should be changed. We have built the tables according to the template provided by the IJERPH (https://www.mdpi.com/files/word-templates/ijerph-template.dot), and the format is in accordance to previous publications in this journal.

  1. - It's not necessary to use in the time "95% CI"

Response: We understand that removing the ‘95%CI’ of the text might make it shorter, but we have opted to keep them, as it is in line with standards in the field. This can also be changed by the journal upon acceptance.

  1. - Throughout the text use the notation β = -16.75, 95% CI -22.12; -11.37. It should be explained the first time it appears in the text.

Response: We have provided an example of the interpretation of the beta coefficient the first time it appeared in the text. Changes were made on lines 180-182.

‘The association between sociodemographic, dietary, and substance use factors with physical activity, sports, and non-sports physical activity is displayed in Table 2, and results were presented as mean differences (β). Females engaged in less total physical activity (β=-16.75, 95%CI -22.12; -11.37; i.e., on average, females engaged in 16.75 fewer minutes of total physical activity than males) ‘

  1. - Regression table 2, 3, and 4.  I would not put the row that is "Reference", it's obviously

Response: Although it is indeed evident which categories are the reference group when analyzing dichotomous variables, this may not be as clear for the Family Structure indicator. We preferred to keep the reference category explicit for ease of interpretation.

  1. The "Discussion" section is correct

Response: Thank you.

  1. The "conclusion" section too short. It must be expanded.

Response: We have included more information about the directions of associations found, as a piece of complementary information that answers the objective of the study. Also, a sentence was added to further highlight the importance of the findings from a public health perspective.

Changes were made on lines 288-299.

‘We found that males engaged in more sports and leisure-time screen time, and less involuntary screen time (i.e., working and studying) than females. Older individuals engaged in more non-sport physical activities and involuntary screen time, and individuals with higher SES engaged in more total physical activity. Higher ingestion of processed food was associated with less total physical activity and increased leisure-time and total screen time, whereas increased consumption of non-processed food was associated with higher participation in sports and total physical activity. Smoking was associated with lower total physical activity, but alcohol and illicit drugs were not associated with any 24-hour movement behavior. Finally, no significant associations were found for sleep duration. These results are relevant for public health, given the importance of engaging in healthy levels of physical activity, sedentary behavior, and sleep. The inclusion of specific sedentary behavior and physical activity types is also novel, as they have different correlates and relate to health in different ways. Efforts should be made to promote physical activity, especially among females, older adolescents, and those at risk of socioeconomic vulnerability, while screen time should also be decreased, especially among males. Promoting healthy eating may also support changes in physical activity and screen time. ‘

References

  1. Perfil Socieconômico Dos Municípios Do Brasil Available online: http://atlasbrasil.org.br/2013/pt/perfil_m/florianopolis_sc (accessed on 18 June 2018).
  2. Knebel, M.; Costa, B.; dos Santos, P.; de Sousa, A.C.; Silva, K. The Conception, Validation, and Reliability of the Questionnaire for Screen Time of Adolescents (QueST). 2020.
  3. Bernardo, M.P.S.L.; Pereira, É.F.; Louzada, F.M.; D’Almeida, V. Sleep Duration in Adolescents of Different Socioeconomic Status. J. Bras. Psiquiatr. 2009, 58, 231–237, doi:10.1590/S0047-20852009000400003.
  4. Schäfer, A.A.; Domingues, M.R.; Dahly, D.L.; Meller, F.O.; Gonçalves, H.; Wehrmeister, F.C.; Assunção, M.C.F. Correlates of Self-Reported Weekday Sleep Duration in Adolescents: The 18-Year Follow-up of the 1993 Pelotas (Brazil) Birth Cohort Study. Sleep Med. 2016, 23, 81–88, doi:10.1016/j.sleep.2016.02.013.
  5. Pereira, É.F.; Barbosa, D.G.; Andrade, R.D.; Claumann, G.S.; Pelegrini, A.; Louzada, F.M.; Pereira, É.F.; Barbosa, D.G.; Andrade, R.D.; Claumann, G.S.; et al. Sleep and Adolescence: How Many Hours Sleep Teenagers Need? J. Bras. Psiquiatr. 2015, 64, 40–44, doi:10.1590/0047-2085000000055.
  6. Buysse, D.J.; Reynolds, C.F.; Monk, T.H.; Berman, S.R.; Kupfer, D.J. The Pittsburgh Sleep Quality Index: A New Instrument for Psychiatric Practice and Research. Psychiatry Res 1989, 28, doi:10.1016/0165-1781(89)90047-4.

Round 2

Reviewer 2 Report

I don't have any comments